# The Prone Lateral Approach for Lumbar Fusion—A Review of the Literature and Case Series

**DOI:** 10.3390/medicina59020251

**Published:** 2023-01-28

**Authors:** Gal Barkay, Ian Wellington, Scott Mallozzi, Hardeep Singh, Isaac L. Moss

**Affiliations:** UConn Musculoskeletal Institute, Department of Orthopaedic Surgery, University of Connecticut School of Medicine, Farmington, CT 06030, USA

**Keywords:** minimally invasive spine surgery, lateral approach, prone lateral, spinal fusion

## Abstract

Lateral lumbar interbody fusion is an evolving procedure in spine surgery allowing for the placement of large interbody devices to achieve indirect decompression of segmental stenosis, deformity correction and high fusion rates through a minimally invasive approach. Traditionally, this technique has been performed in the lateral decubitus position. Many surgeons have adopted simultaneous posterior instrumentation in the lateral position to avoid patient repositioning; however, this technique presents several challenges and limitations. Recently, lateral interbody fusion in the prone position has been gaining in popularity due to the surgeon’s ability to perform simultaneous posterior instrumentation as well as decompression procedures and corrective osteotomies. Furthermore, the prone position allows improved correction of sagittal plane imbalance due to increased lumbar lordosis when prone on most operative tables used for spinal surgery. In this paper, we describe the evolution of the prone lateral approach for interbody fusion and present our experience with this technique. Case examples are included for illustration.

## 1. Introduction

With the evolution of spinal surgery, various approaches to the spine have been utilized in an effort to address a wide array of pathologies in the most efficient manner whilst minimizing patient complications. As a derivative of this, minimally invasive surgical techniques have been growing in popularity. These have been associated with numerous clinical benefits, such as faster postoperative recovery and decreased rates of many complications, thus improving patient experience and reducing costs [1].

In lumbar fusion procedures, access to the anterior column is possible through several approaches, each with its advantages and disadvantages [2]. The lateral transpsoas approach to the lumbar spine, described by Ozgur et al. in 2006 [3], offered a novel, minimally invasive, muscle-sparing approach that provides access to the anterior column. The benefits of this approach have led to its adoption in approximately 20% of lumbar fusion procedures in the United States, according to corporate market research data. Furthermore, the lateral approach has been gaining in popularity in deformity-correction procedures with the increased use of hyperlordotic cages as well as the development of anterior column realignment (ACR) procedures [4,5]. With respect to the classic anterior and posterior approaches to the lateral spine, this approach offers an alternative to address some of the disadvantages associated with traditional approaches. Vascular complications, as well as postoperative ileus and other abdominal complications that are more common in the direct anterior approach, are mostly avoided when employing the lateral approach [2]. Additionally, as an access surgeon is generally not needed, the logistics of scheduling two surgeons for the procedure and the opportunity cost to the approach surgeon, along with the general practice in the American healthcare system of splitting professional fees, which imposes a potential financial burden, are avoided in lateral-approach surgery. The posterior approaches have been criticized for achieving relatively smaller improvements in segmental lordosis and disc height when compared with anterior approaches [6,7,8]. Moreover, the lateral approach offers the ability to insert larger interbody devices that rest on the cortical rim of the vertebral body, thus having the ability to achieve greater indirect decompression with a lower rate of subsidence [9]. Unfortunately, there is still a paucity of data regarding patient outcomes in studies that make direct comparisons between the two approaches [6,9].

The lateral approach has its own disadvantages, the most common being postoperative hip flexor weakness secondary to psoas muscle trauma during the approach. In a review of 451 patients, Lykissas et al. reported a 38% incidence of anterior thigh pain as well as a similar incidence of sensory deficits in the immediate postoperative period. Postoperative motor deficits were reported in 24% of patients. At 18 months follow-up, there was a decrease to 9.6% in sensory and 3.2% in motor symptoms [10]. With the development of advanced neuromonitoring techniques, evidence has shown a decrease in postoperative neurological deficits following lateral surgery [11]. Additional complications reported include rare wound infections, as well as visceral and vascular injury [12].

A crucial disadvantage that has likely hindered the adoption of the lateral approach is the need for patient repositioning. Procedures often necessitate posterior instrumentation and/or decompression in addition to lateral interbody fusion, and, although experienced surgical teams are able to minimize repositioning times, a two-position approach is a cumbersome and costly option for many [13].

The desire to perform lateral as well as posterior procedures whilst maintaining an efficient surgical workflow has brought about different methods of single-position dual-approach surgery. The lateral decubitus single-approach position has been described as a feasible option for lateral interbody fusion as well as posterior instrumentation without the need for patient repositioning [14]. In the lateral position, the anterior column is accessible along the entire length of the lumbar spine, including L5-S1, through ALIF and transpsoas approaches. Posterior instrumentation is then achieved using fluoroscopic or navigated techniques. Several studies have shown no significant differences in postoperative radiographic parameters between single and dual positioning. Furthermore, reduced operative time has been reported [15,16,17,18] in single-position patients as well as reduced fluoroscopy usage, EBL and rate of postoperative ileus. Although relatively efficient for instrumentation techniques, the lateral decubitus position does not allow for additional posterior procedures, such as direct neural decompression or osteotomies, for correction of spinal malalignment. Thus, despite the utility of this approach, its drawbacks have led surgeons to explore additional options for single-position techniques that will allow multidirectional access to the spine.

## 2. The Prone Lateral Approach

In an attempt to overcome the disadvantages of the direct lateral transpsoas approach in the lateral decubitus position, a novel prone lateral approach for access to the lumbar spine was described by Pimenta et al. [19] in 2020 and since then has been gaining in popularity. This approach has several advantages, the main one being the ability to access the anterior column via a direct lateral approach while simultaneously having complete access to the posterior spine for decompression, instrumentation and deformity-correction procedures. Furthermore, this simultaneous access gives the surgeon the flexibility to modify the surgical plan based on parameters achieved with any one aspect of the procedure, such as lordosis correction or the extent of indirect decompression, in any order, without the need to reposition. Additional advantages described with this approach are related to the prone position itself. Prone positioning on a Jackson-type frame allows for increased lordosis and thus improved lumbar sagittal plane correction [20] while instrumenting without additional manual manipulation of the spinal column. This effect can potentially be enhanced with the use of a table with dynamic positioning abilities (e.g., Proaxis, Mizuho OSI, Union City, CA, USA). Furthermore, it has been hypothesized that, while in the prone position, extension of the hips results in a posterior retraction of the psoas muscle. This may move the nerves of the lumbar plexus posteriorly and thus provide a larger safe corridor to access the disc space [21,22], although recent imaging findings have not supported this theory [23]. While it is a relatively new approach, early reports regarding the prone lateral approach have been promising. Lamartina and Berajo reported a non-randomized study of 7 patients undergoing prone lateral interbody fusion and 10 patients undergoing lateral interbody fusion in the lateral decubitus position, demonstrating a shorter operative time in the prone lateral group as well as comparable ODIs and back and leg NRSs at follow-up between the two groups. Pimenta et al. reported a multi-center cohort of 27 patients undergoing prone transpsoas fusion at the L4–5 disc space. They reported that a single patient (4%) had a postoperative motor deficit and that three patients (11%) had postoperative sensory deficits [21], in addition to a neurological complication rate less than that reported in several papers regarding the lateral approach [10,24]. Although promising, this approach has several disadvantages. One example is the inability to reach the anterior column at L5-S1 without repositioning, as can be achieved with a lateral ALIF approach while in the lateral decubitus position [14]. An additional disadvantage secondary to positioning is that the spine cannot easily be laterally flexed in order to enlarge the distance between the caudal ribs and the iliac crest, as is performed in lateral decubitus, although this potential drawback can be partially overcome by specially designed patient positioners. Furthermore, it has been reported that this approach may have an increased learning curve [19,25]. This may be reflected in greater initial preincision times and X-ray times, as reported by Lamartina and Berjano [25]. 

## 3. Our Initial Institutional Experience

Lateral interbody fusion through a prone transpsoas approach is performed by three surgeons at our institution. Since the adoption of the prone lateral approach to the lumbar spine at our institution, this has been our primary method for the achievement of lumbar interbody fusion at the levels above L5-S1. Patient indications for this procedure include fusion for segmental instability and pseudoarthrosis, indirect decompression of foraminal and central stenosis, and the correction of sagittal and/or coronal plane deformity. Each patient is worked up preoperatively with imaging studies to assess for the feasibility of the approach. Standing radiographs are obtained to assess the approach trajectory for indicated levels. The assessment of the height of the iliac crest with regard to the pathological disc space is especially important on X-ray films. A high iliac crest, at or above the L4 pedicle on lateral X-ray, can interfere with the approach to L4–5, requiring consideration of the use of angled instruments to prepare the disc space or consideration of an alternative approach. When treating scoliosis, the approach is most often planned from the concavity, allowing for the correction of coronal plane deformity through positioning, easier access to the cranial and caudal disc spaces, and fewer incisions to treat more levels. Magnetic resonance imaging (MRI) is used to assess the shape and position of the psoas muscle and the lumbar plexus in relation to the disc. An anterior psoas and plexus relative to the disc space is a potential contraindication to performing the direct lateral approach at a given level, as it reduces the likelihood of finding a safe neurological corridor to the disc. The proximity of additional structures, such as major blood vessels, kidneys and the bowel, is assessed via MRI as well. Previous retroperitoneal surgery at the side of the approach is also a relative contraindication, as adhesions may make the blunt dissection to the psoas muscle difficult. When lateral interbody fusion is not feasible, an alternative method of fusion is selected by the surgeon based on their assessment of the surgical goals and the surgical skill set.

Patients treated in our series underwent lateral interbody fusion at one to four levels between L1 and L5 based on the treating surgeon’s assessment of the pathology to be addressed and individual patient anatomies and goals. The surgical approach for prone lateral interbody fusion is similar to that of the classic lateral approach, with several nuances that should be taken into consideration. As compared to the lateral decubitus position, when the patient is in the prone position, gravitational forces pull the rector anteriorly relative to the fixed spine. In addition, the increased lordosis of the lumbar spine when prone on a four-posted Jackson-type frame and thigh extension (as opposed to flexion in the traditional lateral position) result in posterior migration of the psoas muscle relative to the vertebral body. Thus, the posterior aspect of the incision is directly lateral to the posterior border of the neural foramen, as opposed to the posterior border of the disc space. Additionally, the approach corridor is longer in the prone lateral as compared to the lateral decubitus position due to gravitational forces on the abdomen and surrounding soft tissues. In the lateral decubitus position, the surrounding structures shift towards the contralateral side of the patient, making the approach corridor relatively shorter. Thus, the prone lateral approach requires the use of longer retractors and instruments. Our specific patient positioning and surgical technique is similar to that reported by Pimenta et al. [21]. Placement of posterior instrumentation is performed either before or after interbody placement and is accomplished based on the goals of the procedures and surgeon preference, using percutaneous or open approaches employing freehand, fluoroscopically assisted or navigated techniques. If computed-tomography-based navigation techniques are being used, it is important to note that the placement of interbody devices may change the relative position of vertebral bodies and could affect the accuracy of navigation when the scan is obtained prior to interbody instrumentation.

### Patient Selection and Data Collection

All patients who underwent prone lateral interbody fusion at our institution from our adoption of the technique in mid-2020 until March 2022 were assessed. Indications included stenosis requiring disc distraction to achieve foraminal decompression, segmental instability and deformity. Patients with less than 3 months of follow-up were excluded from the analysis. Postoperative complications as well as patient-reported outcomes were collected preoperatively as well at regular postoperative follow-up intervals, including Oswestry Disability Index (ODI), Pain EQ5D, Visual Analogue Scale (VAS) leg and back pain, global report of change (GROC) for pain and function scores as well as patient acceptable symptom state (PASS) scores.

Initial 3-month outcomes of the first 82 patients (133 levels) treated with this approach have been promising. Significant improvements in Oswestry Disability Index (ODI) as well as Pain EQ5D and Visual Analogue Scale (VAS) leg and back pain scores were seen as early as 6 weeks after surgery. At 3 months, 87% of patients treated with the prone transpsoas approach at our institution reported their pain as being at least “a little bit better” than their preoperative levels, with 73% reporting that they were “quite a bit better” at 3 months. 

Complications following lateral interbody fusion via a prone technique at our institution were similar to those seen following laterally positioned surgery. At least one postoperative complication related to the index procedure occurred in 76% of our patients. The most common complication was ipsilateral hip flexor pain, with 45% of patients reporting this complication. This is to be expected given the retraction of the psoas muscle intraoperatively and may be considered a side effect of treatment, as opposed to a true complication. Ipsilateral hip flexor weakness was observed in 39% of patients. Individual-level retractor time was not recorded in this series. Additionally, there were two cases (2.4%) of postoperative femoral nerve palsy, both of which resolved spontaneously within 3 months, and 3 cases (3.7%) of inadvertent anterior longitudinal ligament rupture, which were treated intraoperatively with lateral plating. There were no patients with permanent femoral nerve palsy and no visceral or major vascular injuries in our series. Ultimately, the reported high complication rate is related predominantly to the high rate of hip flexor pain and weakness following this procedure. This is an expected outcome which patients are extensively counseled about preoperatively and resolves in roughly 50% of patients at 3 months and in 90% of patients at 1 year [26]. A noteworthy point is that this series included surgeon learning curves, which could possibly explain the relatively high complication rates found.

## 4. Case Examples

### 4.1. Case 1

A 58-year-old female presented with a history of significant mechanical back pain as well as bilateral leg pain, worst in her right foot. Imaging revealed a facet cyst causing lateral recess stenosis at L3–4 with a grade-2 lytic spondylolisthesis at L4–5 causing severe neuroforaminal stenosis (Figure 1A–D). First, the L3–4 facet cyst was addressed with a minimally invasive tubular posterior decompression, medial facetectomy and resection of the facet cyst. This was followed by percutaneous instrumentation of L4 and L5 with pedicle screws. The listhesis was then partially reduced by the placement of rods with set caps left untightened to allow for additional reduction and distraction when inserting the interbody cage (Figure 1E). With the patient still in the prone position, a lateral incision was made, the L4–5 disc space was exposed through the transpsoas approach and preparation of the disc space was performed. Following this, the cage was placed through the lateral approach to achieve further reduction of listhesis and indirect decompression. Set caps were then final-tightened, resulting in a near-complete reduction of the listhesis, a significant increase in foraminal height and the restoration of segmental lordosis (Figure 1E,F). The patient was discharged home on POD 1 with complete resolution of radiculopathy. This result has remained durable at 1-year postoperative follow-up.

### 4.2. Case 2

A 65-year-old female presented with axial back pain with bilateral leg pain, most significant in her left anterior thigh. Imaging demonstrated sagittal imbalance with a lumbar lordosis/pelvic incidence mismatch of approximately 30 degrees. Coronal imbalance was also present with segmental scoliosis at L2–4, resulting in foraminal stenosis at these levels and a left trunk shift of 45 mm. No central stenosis was noted on imaging. The patient was indicated for L1 to L5 lateral interbody fusion with percutaneous posterior instrumented fusion. In the operating room, following prone positioning on a Jackson-type frame, lumbar lordosis improved by approximately 10 degrees. Percutaneous pedicle instrumentation was performed with augmented-reality navigation assistance. The disc spaces were then approached through the concavity of the scoliosis on the left side via a lateral transpsoas approach. Following disc preparation, 15° lordotic interbody cages were inserted. Sagittal as well as coronal plane correction were achieved with a lumbar lordosis and a pelvic incidence of approximately 60 degrees (Figure 2). The patient was discharged on POD 4.

## 5. Future Directions

Although the prone lateral transpsoas approach is extremely promising, it is a relatively new technique that has so far been adopted by relatively few spine surgeons. Due to this, additional research is warranted to further establish its utility and safety, as well as the associated learning curve. Studies with long-term follow-up examining outcomes with respect to approaches in the lateral decubitus position, as well as posterior approaches, should be conducted to better establish differences between the various approaches to lumbar fusion. Additionally, specific focus should be aimed at the reduction of the possible neurological sequelae of this procedure—an issue that deters many surgeons from adopting it. Possible methods to address this issue could be the development of techniques for the reduction of retractor time, hastening the surgeon’s learning curve through advanced teaching methods, and the possible administration of local or systemic drugs that could act to modulate neurogenic symptoms. Furthermore, more advanced neuromonitoring techniques should be developed in order to provide the surgeon with a more precise real-time assessment of possible insult to the lumbar plexus. Future developments are also needed to address the weaknesses of many current systems, such as improving the working space between the ribcage and the iliac crest through better patient positioning devices. The introduction of enabling technologies is also likely to have a beneficial effect on the evolution and adoption of this method. Navigation and augmented-reality technologies have the potential to aid retractor docking and access, as well as hardware placement, in a more precise and protected manner with less radiation exposure. Furthermore, real-time spatial localization could aid in additional applications for prone lateral techniques, such as the treatment of tumors and infections.

## 6. Conclusions

The prone lateral approach to the spine offers a versatile and powerful approach for spinal fusion and deformity-correction procedures. We anticipate further adoption of this technique with additional applications as it continues to evolve.

## Figures and Tables

**Figure 1 medicina-59-00251-f001:**
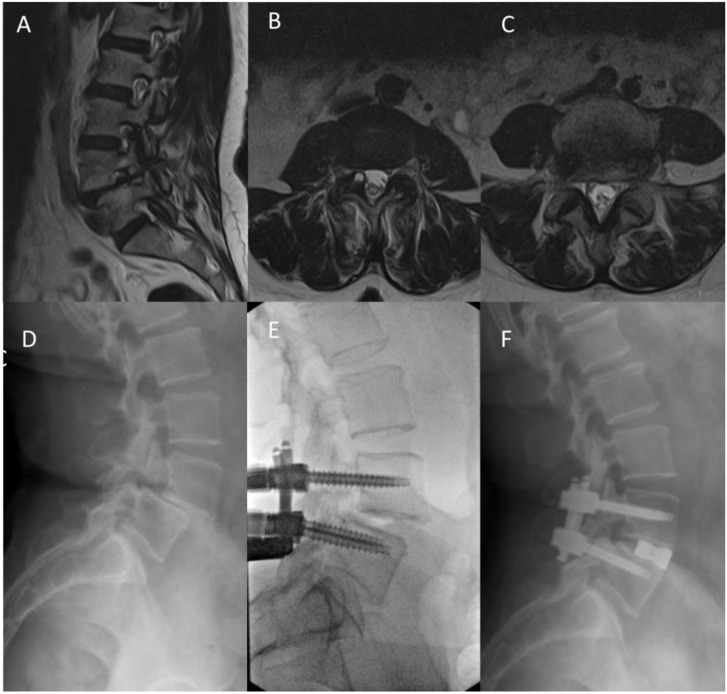
Case 1: (**A**) parasagittal T2 MRI image demonstrating severe foraminal stenosis; (**B**) axial T2 MRI image of L3–4 demonstrating rt facet cyst; (**C**) axial T2 MRI image of L4–5 demonstrating foraminal stenosis; (**D**) preoperative lateral X-ray demonstrating grade-2 lytic spondylolisthesis at L4–5; (**E**) intraoperative fluoroscopy demonstrating partial reduction following posterior instrumentation in the prone position; (**F**) postoperative lateral X-ray demonstrating posterior instrumentation and anterior interbody at L4–5 with complete reduction of lysthesis and widening of the neuroforamina.

**Figure 2 medicina-59-00251-f002:**
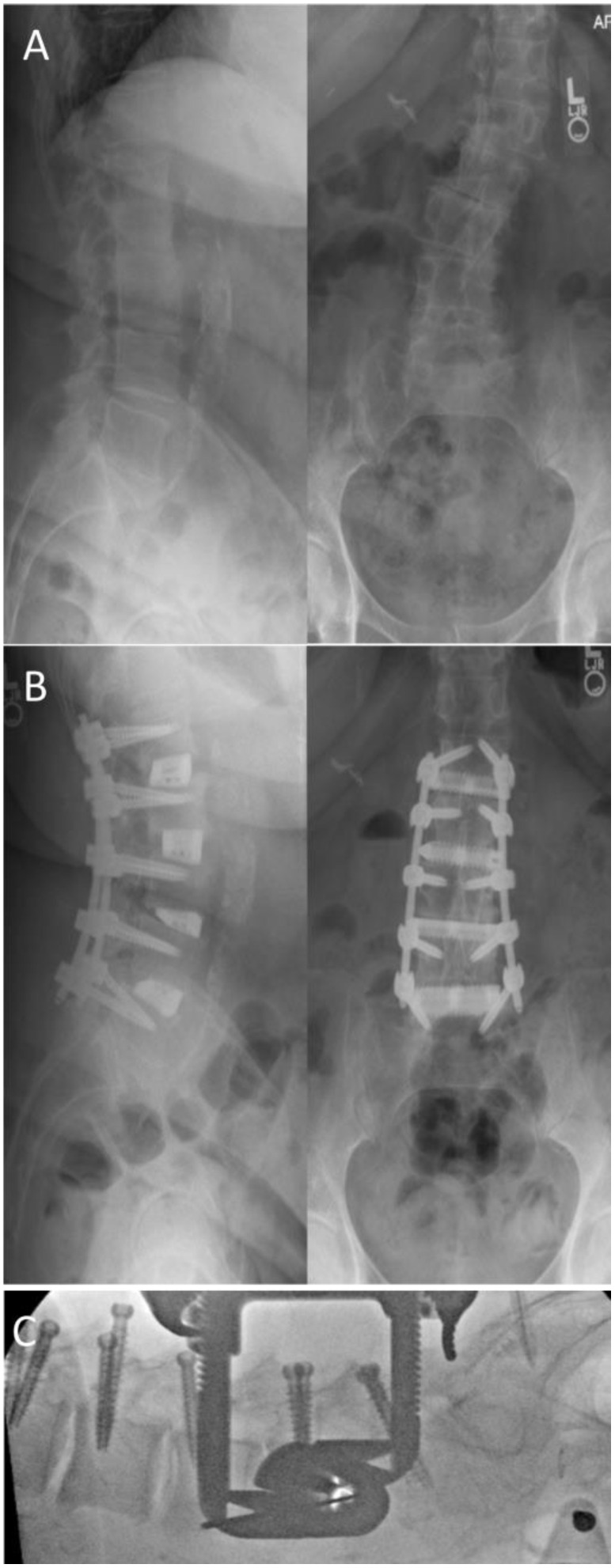
Case 2: (**A**) preoperative AP and lateral X-rays demonstrating sagittal and coronal imbalance; (**B**) postoperative AP and lateral images demonstrating posterior fixation and interbody cages as L1-L5 with correction of sagittal and coronal malalignment; (**C**) intraoperative lateral fluoroscopy demonstrating partial correction of lordosis following prone positioning.

## Data Availability

Further data for this study is not available to the public.

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
