# Peer review of "The Prone Lateral Approach for Lumbar Fusion—A Review of the Literature and Case Series"

_medicina, 2023, doi:10.3390/medicina59020251_

Round 1
Reviewer 1 Report
Thank you for providing a narrative review of the history of lateral lumbar interbody fusions and the adoption of the prone lateral approach. It was a good read but I believe it would benefit from some additional information in your case series. Please see my comments below. Thank you for the opportunity to review this manuscript.
“The benefits of this approach have led to its adoption in approximately 20% of lumbar fusion procedures in the United States – needs citation. Most databases are unable to clarify whether anterior lumbar fusions are ALIF or LLIFs since they are coded the same.
Additionally, the need for an access surgeon is a logistical and financial burden which is avoided with the lateral approach” Please describe how the access surgeon is a financial burden. Less reimbursement for the spine surgeon or greater reimbursement rates to pay the spine and access surgeon?
The posterior approaches have been criticized for relatively smaller improvements in segmental lordosis and disc height when compared with anterior approaches(6),(7),(8). While this is true it does not appear to negatively affect patient outcomes. I think this needs to be added since we are not treating radiographs.
In order to discuss your initial patient experience a methodology section is required. Did you use any specific positioning techniques? Do you always instrument posteriorly prior to the placement of the interbody cage? It appears all LLIFs were not done at the same vertebral level and some were multilevels-please elaborate on your patient population? What were your indications/preoperative diagnoses for LLIFs and how does your practice determine which fusion technique to use (ALIF/LLIF/TLIF, etc?) What were the preop and postop PROs? Whose publication(s) did you use to define the MCID? How long was the average retractor time in your case series and did this differ based on the use of augmented reality or other technologies? Did differences in retractor time length lead to greater postoperative complications? What was minimum follow up time of your patients? Did any patient have permanent complications? These are all important questions to answer in your methodology and results section and I think some of these are also questions that need to be answered in future research. Please expand upon these points in your future directions section.
Author Response
We thank the reviewer for the comments regarding our manuscript. these are addressed below.
“The benefits of this approach have led to its adoption in approximately 20% of lumbar fusion procedures in the United States – needs citation. Most databases are unable to clarify whether anterior lumbar fusions are ALIF or LLIFs since they are coded the same.
- The manuscript has been revised in accordance with the reviewer's remark.
- Page 3 lines 12-14: The benefits of this approach have led to its adoption in approximately 20% of lumbar fusion procedures in the United States, according to corporate market research data.
Additionally, the need for an access surgeon is a logistical and financial burden which is avoided with the lateral approach” Please describe how the access surgeon is a financial burden. Less reimbursement for the spine surgeon or greater reimbursement rates to pay the spine and access surgeon?
- The manuscript has been revised in accordance with the reviewer's remark.
- Page 3 lines 20-24: Additionally as an access surgeon is generally not needed, the logistics of scheduling two surgeons for the procedure and the opportunity cost to the approach surgeon along with the general practice in the American healthcare system of splitting professional fees, posing a potential financial burden, that is avoided in lateral approach surgery.
The posterior approaches have been criticized for relatively smaller improvements in segmental lordosis and disc height when compared with anterior approaches(6),(7),(8). While this is true it does not appear to negatively affect patient outcomes. I think this needs to be added since we are not treating radiographs.
- The manuscript has been revised in accordance with the reviewer's remark.
- Page 3 lines 28-30: Unfortunately, there is still a paucity of data regarding patients outcomes in studies with direct comparisons between the two approaches(6,9).
In order to discuss your initial patient experience a methodology section is required.
- The manuscript has been revised in accordance with the reviewer's remark.
- Page 6 lines 4-12: Patient Selection and Data Collection. All patients who underwent prone lateral interbody fusion at our institution from our adoption of the technique in mid 2020 until March 2022 were assessed. Indications included stenosis requiring disc distraction to achieve foraminal decompression, segmental instability and deformity. Patients with less than 3 months of follow up were excluded from the analysis. Postoperative complications as well as patient reported outcomes were collected pre-operatively as well at regular postoperative follow up intervals, including: Oswestry Disability Index (ODI), Pain EQ5D,Visual Analogue Scale (VAS) leg and back pain, global report of change (GROC) for pain and function scores as well as patient acceptable symptoms state (PASS) scores.
Did you use any specific positioning techniques?
- The manuscript has been revised in accordance with the reviewer's remark.
- Page 5 lines 41-42: Our specific patient positioning and surgical technique is similar to that reported by Pimenta et al.(21).
Do you always instrument posteriorly prior to the placement of the interbody cage?
- The manuscript has been revised in accordance with the reviewer's remark.
- Page 5 line 42 to page 6 line 2: Placement of posterior instrumentation is performed either before or after interbody placement and is accomplished based on the goals of the procedures and surgeon preference, using percutaneous or open approaches employing freehand, fluoroscopically-assisted or navigated techniques. If computed-tomography-based navigation techniques are being used, it is important to note that placement of interbody devices may change the relative position of vertebral bodies and could affect accuracy of navigation when the scan is obtained prior to interbody instrumentation.
It appears all LLIFs were not done at the same vertebral level and some were multilevels-please elaborate on your patient population? What were your indications/preoperative diagnoses for LLIFs and how does your practice determine which fusion technique to use (ALIF/LLIF/TLIF, etc?)
- The manuscript has been revised in accordance with the reviewer's remarks.
- Page 5 lines 6-29: Since the adoption of the prone lateral approach to the lumbar spine at our institution, this has been our primary method for the achievement of lumbar interbody fusion at the levels above L5-S1. Patient indications for this procedure include fusion for segmental instability and pseudoarthrosis, indirect decompression of foraminal and central stenosis, and the correction of sagittal and/or coronal plane deformity. Each patient is worked up preoperatively with imaging studies to assess for the feasibility of the approach. Standing radiographs are obtained to assess the approach trajectory for indicated levels. The assessment of the height of the iliac crest with regard to the pathologic disc space is especially important on x-ray films. A high iliac crest, at or above the L4 pedicle on lateral x-ray can interfere with the approach to L4-5, requiring consideration for the use of angled instruments to prepare the disc space or consideration of an alternative approach. When treating scoliosis, the approach is most often planned from the concavity, allowing for correction of the coronal plane deformity through positioning, easier access to the cranial an caudal disc spaces and fewer incisions to treat more levels. Magnetic Resonance Imaging (MRI) is obtained to assess the shape and position of the psoas muscle and the lumbar plexus with relation to the disc. An anterior psoas and plexus, relative to the disc space is a potential contraindication to performing the direct lateral approach at a given level, as it reduces the likelihood of finding a safe neurologic corridor to the disc. Proximity of additional structures such as major blood vessels, kidney and bowel is assessed on MRI as well. Previous retroperitoneal surgery of the side of the approach, is also a relative contraindication as adhesions may make the blunt dissection to the psoas muscle difficult. When lateral interbody fusion is not feasible, an alternative method of fusion is selected by the surgeon based on their assessment of the surgical goals and surgical skill set.
Patients treated in our series underwent lateral interbody fusion at one to four levels between L1 and L5, based on the treating surgeon’s assessment of the pathology needed to be addressed and individual patient anatomies and goals.
- Page 5 lines 6-29: Since the adoption of the prone lateral approach to the lumbar spine at our institution, this has been our primary method for the achievement of lumbar interbody fusion at the levels above L5-S1. Patient indications for this procedure include fusion for segmental instability and pseudoarthrosis, indirect decompression of foraminal and central stenosis, and the correction of sagittal and/or coronal plane deformity. Each patient is worked up preoperatively with imaging studies to assess for the feasibility of the approach. Standing radiographs are obtained to assess the approach trajectory for indicated levels. The assessment of the height of the iliac crest with regard to the pathologic disc space is especially important on x-ray films. A high iliac crest, at or above the L4 pedicle on lateral x-ray can interfere with the approach to L4-5, requiring consideration for the use of angled instruments to prepare the disc space or consideration of an alternative approach. When treating scoliosis, the approach is most often planned from the concavity, allowing for correction of the coronal plane deformity through positioning, easier access to the cranial an caudal disc spaces and fewer incisions to treat more levels. Magnetic Resonance Imaging (MRI) is obtained to assess the shape and position of the psoas muscle and the lumbar plexus with relation to the disc. An anterior psoas and plexus, relative to the disc space is a potential contraindication to performing the direct lateral approach at a given level, as it reduces the likelihood of finding a safe neurologic corridor to the disc. Proximity of additional structures such as major blood vessels, kidney and bowel is assessed on MRI as well. Previous retroperitoneal surgery of the side of the approach, is also a relative contraindication as adhesions may make the blunt dissection to the psoas muscle difficult. When lateral interbody fusion is not feasible, an alternative method of fusion is selected by the surgeon based on their assessment of the surgical goals and surgical skill set.
What were the preop and postop PROs?
- The manuscript has been revised in accordance with the reviewer's remark.
- Page 6 lines 9-12: patient reported outcomes were collected pre-operatively as well at regular postoperative follow up intervals, including: Oswestry Disability Index (ODI), Pain EQ5D,Visual Analogue Scale (VAS) leg and back pain, global report of change (GROC) for pain and function scores as well as patient acceptable symptoms state (PASS) scores.
Whose publication(s) did you use to define the MCID?
- The manuscript has been revised in accordance with the reviewer's remark. The use of the term "MCID" was done without the proper reference. This was done unintentionally. We have revised our manuscript with correction of the text to the data that was collected:
- Page 6 lines 15-18: At 3 months, 87% of patients treated with the prone transpsoas approach at our institution reported their pain as being at least “a little bit better” than their preoperative levels, with 73% reporting that they were “quite a bit better” at 3 months.
How long was the average retractor time in your case series and did this differ based on the use of augmented reality or other technologies? Did differences in retractor time length lead to greater postoperative complications?
- The manuscript has been revised in accordance with the reviewer's remark.
- Page 6 line 25: Individual level retractor time was not recorded in this series.
What was minimum follow up time of your patients?
- The manuscript has been revised in accordance with the reviewer's remark.
- Page 6 line 7-8: Patients with less than 3 months of follow up were excluded from the analysis.
Did any patient have permanent complications?
- The manuscript has been revised in accordance with the reviewer's remark.
- Page 6 line 28-29: There were no patients with permanent femoral nerve palsy and no visceral or major vascular injuries in our series.
I think some of these are also questions that need to be answered in future research. Please expand upon these points in your future directions section.
- The manuscript has been revised in accordance with the reviewer's remark.
- Page 7 line 26-35: Studies with long term follow up examining outcomes with respect to approaches in the lateral decubitus position, as well as posterior approaches should be conducted to better establish differences between the various approaches to lumbar fusion. Additionally, specific focus should be aimed towards the reduction of the possible neurological sequelae of this procedure, an issue that deters many surgeons from adopting it. Possible methods to address this issue could be development of techniques for the reduction of retractor time, hastening of the surgeon’s learning curve through advanced teaching methods and the study of possible administration of local or systemic drugs that could act in the modulation of neurogenic symptoms. Furthermore, more advanced neuromonitoring techniques should be developed in order to provide the surgeon with a more precise real time assessment of possible insult to the lumbar plexus.
Reviewer 2 Report
Authors describe a very interesting new surgical tecniqye: the lateral approach for lumbar fusion. Further cases should be studied but this approach apperars safe and minimally invasive for patients.
Should authors provide some follow-up data?
Author Response
We thank the reviewer for the comments regarding our manuscript. these are addressed below.
Further cases should be studied but this approach apperars safe and minimally invasive for patients.
- We thank the reviewer for this remark. We intend to continue the study of patients at institution in order to better understand our practice as well as to provide evidence to the scientific community.
Should authors provide some follow-up data?
- We thank the reviewer for this remark. As stated in page 6 lines 7-8: "Patients with less than 3 months of follow up were excluded from the analysis.". Further revision of the manuscript has been done in accordance with this remark:
- Page 7 lines 26-28: . Studies with long term follow up examining outcomes with respect to approaches in the lateral decubitus position, as well as posterior approaches should be conducted to better establish differences between the various approaches to lumbar fusion.
Reviewer 3 Report
Dear authors
Being able to insert a lateral interbody cage from the prone position is indeed a relatively new idea that is gaining traction. It would be interesting if you could provide more details on your technique and the variations in the corridor anatomy when comparing a lateral position vs a prone position. Your technique appears to have a higher rate of hip flexor pain compared to the traditional lateral positioning. Could this be due to the initial learning curve that you had that requires more excessive retraction for this approach or, as you mentioned in the paper, being in prone position requires more retraction just due to the position? Could you recommend some steps/techniques that can be done for readers who would be interested to adopt your technique?
It would also be interesting to look at long term results as well as comparison to a completely posterior approach technique. Would you be able to recommend certain scenarios when your technique would be superior/more preferable compared to a completely posterior approach technique?
Thank you
Author Response
We thank the reviewer for the comments regarding our manuscript. these are addressed below.
It would be interesting if you could provide more details on your technique and the variations in the corridor anatomy when comparing a lateral position vs a prone position.
- The manuscript has been revised in accordance with the reviewer's remark.
- Page 5 lines 29-41: . The surgical approach for prone lateral interbody fusion is similar to that of the classic lateral approach with several nuances that should be taken into consideration. As compared to the lateral decubitus position, when the patient is in the prone position, gravitational forces pull the rector anteriorly relative to the fixed spine. In addition, the increased lordosis of the lumbar spine when prone on a 4-posted Jackson-type frame and thigh extension (as opposed to flexion in the traditional lateral position, result in posterior migration of the psoas muscle relative to the vertebral body. Thus, the posterior aspect of the incision is directly lateral to the posterior border of the neural foramen, as opposed to the posterior border of the disc space. Additionally, the approach corridor is longer in the prone lateral as compared to the lateral decubitus position, due to gravitational forces on the abdomen and surrounding soft tissues. In the lateral decubitus position the surrounding structures shift towards the contralateral side of the patient making the approach corridor relatively shorter. Thus, the prone lateral approach requires the use of longer retractors and instruments.
Your technique appears to have a higher rate of hip flexor pain compared to the traditional lateral positioning. Could this be due to the initial learning curve that you had that requires more excessive retraction for this approach or, as you mentioned in the paper, being in prone position requires more retraction just due to the position?
- The manuscript has been revised in accordance with the reviewer's remark.
- Page 6 lines 32-34: A noteworthy point is that this series included the surgeons’ learning curve, this could possibly explain the relatively high complication rates found.
Could you recommend some steps/techniques that can be done for readers who would be interested to adopt your technique?
- The manuscript has been revised in accordance with the reviewer's remark.
- Page 5 line 29 to page 6 line 2: The surgical approach for prone lateral interbody fusion is similar to that of the classic lateral approach with several nuances that should be taken into consideration. As compared to the lateral decubitus position, when the patient is in the prone position, gravitational forces pull the rector anteriorly relative to the fixed spine. In addition, the increased lordosis of the lumbar spine when prone on a 4-posted Jackson-type frame and thigh extension (as opposed to flexion in the traditional lateral position, result in posterior migration of the psoas muscle relative to the vertebral body. Thus, the posterior aspect of the incision is directly lateral to the posterior border of the neural foramen, as opposed to the posterior border of the disc space. Additionally, the approach corridor is longer in the prone lateral as compared to the lateral decubitus position, due to gravitational forces on the abdomen and surrounding soft tissues. In the lateral decubitus position the surrounding structures shift towards the contralateral side of the patient making the approach corridor relatively shorter. Thus, the prone lateral approach requires the use of longer retractors and instruments. Our specific patient positioning and surgical technique is similar to that reported by Pimenta et al.(21). Placement of posterior instrumentation is performed either before or after interbody placement and is accomplished based on the goals of the procedures and surgeon preference, using percutaneous or open approaches employing freehand, fluoroscopically-assisted or navigated techniques. If computed-tomography-based navigation techniques are being used, it is important to note that placement of interbody devices may change the relative position of vertebral bodies and could affect accuracy of navigation when the scan is obtained prior to interbody instrumentation.
It would also be interesting to look at long term results as well as comparison to a completely posterior approach technique.
- The manuscript has been revised in accordance with the reviewer's remark.
- Page 7 line 26-28: Studies with long term follow up examining outcomes with respect to approaches in the lateral decubitus position, as well as posterior approaches should be conducted to better establish differences between the various approaches to lumbar fusion.
Would you be able to recommend certain scenarios when your technique would be superior/more preferable compared to a completely posterior approach technique?
- The manuscript has been revised in accordance with the reviewer's remark.
- Page 3 lines 24-30: The posterior approaches have been criticized for relatively smaller improvements in segmental lordosis and disc height when compared with anterior approaches(6),(7),(8). Moreover, the lateral approach offers the ability to insert larger interbody devices that rest on the cortical rim of the vertebral body, thus having the ability to achieve greater indirect decompression with a lower rate of subsidence(9). Unfortunately, there is still a paucity of data regarding patients outcomes in studies with direct comparisons between the two approaches(6,9).
Round 2
Reviewer 1 Report
The authors addressed all of my comments during their revisions. I believe the manuscript is sufficiently improved and I have no additional comments to provide.